# Characterising the interventions designed to affect the reporting of musculoskeletal imaging: a scoping review protocol using the COM-B model

Edward Kirby ![ORCID],[1] Andrew MacMillan,[2] Bernard X W Liew ![ORCID],[3] Andrew Brinkley,[3] Andrew Bateman[4]

[1]MSK Physiotherapy Dept, Essex Partnership University NHS Foundation Trust, Wickford, UK
[2]Research department, University College of Osteopathy, London, UK
[3]School of Sport, Rehabilitation and Exercise Sciences, University of Essex, Colchester, UK
[4]School of Health and Social Care, University of Essex, Colchester, UK

**Correspondence to**
Edward Kirby;
ed@edkirbyphysio.co.uk

## ABSTRACT

**Introduction** Attributing musculoskeletal (MSK) pain to normal and commonly occurring imaging findings, such as tendon, cartilage and spinal disc degeneration, has been shown to increase people's fear of movement, reduce their optimism about recovery and increase healthcare costs. Interventions seeking to reduce the negative effects of MSK imaging reporting have had little effect. To understand the ineffectiveness of these interventions, this study seeks to scope their behavioural targets, intended mechanisms of action and theoretical underpinnings. This information alongside known barriers to helpful reporting can enable researchers to refine or create new more targeted interventions.

**Methods and analysis** The scoping review will be conducted in accordance with the JBI methodology for scoping reviews and the Preferred Reporting Items for Systematic Reviews and Meta-Analyses extension for Scoping Reviews. Search terms will be devised by the research team. Searches of MEDLINE, EMBASE, CINAHL, AMED and PsycINFO from inception to current day will be performed. The review will include studies, which have developed or evaluated interventions targeting the reporting of MSK imaging. Studies targeting the diagnosis of serious causes of MSK pain will be excluded. Two independent authors will extract study participant data using predefined extraction templates and intervention details using the Template for Intervention Description and Replication checklist. Interventions will be coded and mapped to the technique, mechanism of action and behavioural target according to the Capability, Opportunity, Motivation-Behaviour (COM-B) model categories. Any explicit models or theories used to inform the selection of interventions will be extracted and coded. The study characteristics, behaviour change techniques identified, behavioural targets according to the COM-B and context specific theories within the studies will be presented in narrative and table form.

**Ethics and dissemination** The information from this review will be used to inform an intervention design process seeking to improve the communication of imaging results. The results will also be disseminated through a peer-reviewed publication, conference presentations and stakeholder events.

## STRENGTHS AND LIMITATIONS OF THIS STUDY

⇒ Novel use of a validated behaviour change framework to retrospectively highlight the targets of existing interventions.
⇒ Comprehensive search strategy using a range of databases, handsearching and rigorous article screening performed by two independent authors and a third author to reconcile discrepancies.
⇒ The communication of imaging reporting may be an embedded component of broader treatment approaches, and therefore, difficult to identify during literature searching.
⇒ The quality of the coding may be limited by the accuracy of the intervention reporting.

## INTRODUCTION
### Rationale

Data from the Global Burden of Disease Study indicate that musculoskeletal (MSK) disorders (low back pain, neck pain, osteoarthritis, rheumatoid arthritis and other MSK disorders) are the leading cause of disability globally in 2019.[1] MSK conditions such as arthritis and back pain affected an estimated 18.8 million people across the UK in 2017 and accounted for more than 22% of the total burden of ill health in the UK with an estimated cost to the National Health Service of over £5 billion annually.[2] Despite guidance on the management of the most common MSK conditions,[3 4] studies have shown that in some cases healthcare practitioners (HCPs) may negatively affect their patients' disorder by influencing or perpetuating negative or unhelpful health beliefs.[5]

Prior qualitative studies have identified that patients' primary goal for attending health services for MSK pain is a desire to identify a clear pathoanatomical cause[6] with the added belief that MSK imaging is a useful test to discover the source of the pain.[7] This may seem coherent based on the prevailing biomedical model, which determines that visible abnormalities

on imaging are causally linked with pain.[8] Epidemiological studies, however, have found that abnormalities, such as disc degeneration in the spine,[9] degenerative cartilage tears in the knee[10] and tendon tears in the shoulder[11] are equally common in asymptomatic populations. The tenuous association between many common MSK imaging findings and pain means that HCPs have difficulty confidently providing a specific diagnosis to patients in such cases.

The perception held by some HCPs that their patients desire a clear diagnosis, combined with difficulty understanding reports and communicating diagnostic uncertainty have been cited as reasons for HCPs continuing to attribute the cause of MSK pain to findings of doubtful significance on imaging.[6] A number of negative consequences from speculative diagnostic labelling such as 'wear and tear', 'damage' and 'degeneration' have been reported. These include: increased fear of movement,[5] perception of poorer prognosis,[12] reduced patient confidence in conservative management,[13] withdrawal from valued life activities,[14] impaired general health outcomes[15] and increases in unnecessary or low value treatments such as injections and surgery.[12 16]

In an attempt to prevent harmful MSK image reporting behaviours, studies have evaluated the withholding of imaging reports,[15] reassurance about the lack of serious pathology[17] and the insertion of information about the prevalence of findings in the pain free population.[18] A recent systematic review, however, found little evidence of effectiveness for any of the interventions.[19] While these interventions attempted to change clinician-reporting behaviours, they did not outline a thorough intervention design process informed by theory. The National Institute for Health and Care Research and Medical Research Council[20] recommend using behavioural science in the development of interventions to improve effectiveness and the lack of such a process in these studies may have contributed to their lack of effectiveness.

One approach to developing an effective behaviour change intervention for MSK reporting would be to use a framework such as the Behaviour Change Wheel (BCW).[21] The BCW outlines a process which uses the Capability, Opportunity, Motivation-Behaviour (COM-B) model to identify what needs to change and then selects behaviour change techniques (BCT's) based on this analysis.[21] The COM-B model can also be used as a framework to retrospectively highlight the behavioural targets of prior interventions. For example, Hall et al[22] analysed existing interventions to ascertain which BCTs[23] have been used to improve adherence to evidence-based low back pain imaging requests and found that the majority of included studies 'lacked the use of any theory or framework to inform their intervention design'. By using the COM-B model as a framework to map the interventions seeking to affect imaging guideline adherence, they found that the interventions 'failed to target known physician-reported barriers to following LBP imaging guidelines'.

Using the COM-B model to map existing MSK imaging reporting interventions to their behavioural targets in this way, will elucidate the congruence of existing interventions with the known barriers and enablers of helpful

reporting. This information can enable researchers to modify, adapt or develop new and potentially more targeted and effective reporting interventions.

A preliminary search of PROSPERO, Open Science Framework, MEDLINE, the Cochrane Database of Systematic Reviews and the JBI Evidence Synthesis indicates that no study has attempted to map the BCTs used or the theory underpinning interventions designed to affect the communication of MSK imaging findings.

## Objectives

The objective of this review is to discover which BCTs have been employed, what the behavioural targets were and which behavioural theories and theoretical models underpinned the interventions designed to affect MSK image reporting. This information may provide direction for the design of future interventions, by highlighting strategies that show promise and strategies that have not been explored.

## METHODS

The proposed scoping review will be conducted in accordance with the JBI methodology for scoping reviews[24] and the Preferred Reporting Items for Systematic Reviews and Meta-Analyses extension for Scoping Reviews (PRISMA).[25] The protocol was registered with the Open Science Framework https://doi.org/10.17605/OSF.IO/ECYS8. The planned start date for the study is 17 July and is planned to end on October 2023. Any deviations from the protocol will be reported and justified in the methods section of the final review manuscript.

### Eligibility criteria

The review will include studies that have developed or evaluated interventions to target the communication of MSK imaging findings. Studies in any healthcare setting worldwide and published in any language will be included, where languages other than English will be translated. This review will include not limit studies by publication year. Studies that include multiple interventions or broad treatment approaches will be included if it was possible to isolate specific BCTs intended to affect the communication of imaging findings within them. Studies using qualitative methods to develop interventions will be included if their findings identify a target for interventions. The review will exclude studies focusing on serious or specific known causes of MSK pain such as fracture, malignancy, infection and inflammatory arthritis.

This scoping review will consider both experimental and quasi-experimental study designs including randomised controlled trials, non-randomised controlled trials, before-and-after studies and interrupted time-series studies, cluster randomised trials, non-randomised cluster trials, controlled and uncontrolled before-and-after studies and cross-sectional studies.

### Information sources and search strategy

The search strategy will aim to locate published studies. A three-step search strategy will be used in this review. First,

an initial limited search of MEDLINE (PubMed) and CINAHL (EBSCO) was undertaken to identify articles on the topic. The text words contained in the titles and abstracts of relevant articles, and the index terms used to describe the articles will be used to develop a full search strategy for MEDLINE, EMBASE, CINAHL, AMED and PsycINFO (see online supplemental appendix 1). The search will be performed in July 2023. We will email experts in the field of imaging and LBP to identify any studies that may have been missed by the search. The search strategy, including all identified keywords and index terms, will be adapted for each database and/or information source. The reference list of all included sources of evidence will be screened for additional studies. A specialist healthcare librarian will review the search strategy.

## Study selection

Following the search, all identified citations will be collated and uploaded into Refworks (Proquest, Michigan, USA) and duplicates removed. Following a pilot test, titles and abstracts will then be screened by two or more independent reviewers for assessment against the inclusion criteria for the review. Potentially relevant sources will be retrieved in full and assessed in detail against the inclusion criteria by two or more independent reviewers. Reasons for exclusion of papers at full text that do not meet the inclusion criteria will be recorded and reported in the systematic review. Any disagreements that arise between the reviewers at each stage of the selection process will be resolved through discussion, or with an additional reviewer/s. The results of the search and the study inclusion process will be reported in full in the final systematic review and presented in a PRISMA flow diagram[26]

## Assessment of methodological quality

As the objective of the study is to determine which BCTs have been employed, what the behavioural targets were and which behavioural theories and theoretical models underpinned the interventions designed to affect MSK image reporting, the outcomes and, therefore, assessment of methodological quality is not considered necessary and will not be performed.

## Data extraction

Two reviewers will independently extract the study characteristics (year, country, setting, design, patient numbers (n), MSK area, outcome measures, effect sizes barriers, enablers, confounders and modifiers) and intervention information using the Template for Intervention Description and Replication (TIDieR).[27]

The TIDieR template will be used to characterise the intervention. A draft coding manual will be created based on the BCT taxonomy v1 (see online supplemental appendix 2) and piloted. Coding will be based on principles outlined in Lorencatto *et al*.[28] The draft data extraction tool will be modified and revised as necessary during the process of extracting data from each included evidence source. Any coding disagreements that arise between the reviewers will be resolved through discussion, or with a third reviewer. Authors of papers will be contacted to request missing or additional data, where required.

The Theory and Techniques Tool (TATT) (available at: https://theoryandtechniquetool.humanbehaviourchange.org/tool) will be used to map BCTs to their corresponding mechanisms of action in the TDF. The TATT is a map of 74 BCTs linking to 26 mechanisms of action and attributing strength to the association based on research and expert consensus.

Theories explicitly mentioned that either inform the intervention or where the intervention tests or creates the theory will be extracted. Item 2 on the Tidier template— 'why: describe the use of any rationale, theory or goal of the elements essential to the intervention' will be used to extract both evidence of behavioural theory used and context specific models. The Painter criteria[29] will then be applied to the details extracted. This categorisation (see online supplemental appendix 3) distinguishes the use of theory into the following categories: (1) informed by theory, (2) applied theory, (3) testing theory or (4) building or creating theory.

## Data presentation

The study characteristics, BCTs identified, behavioural targets (according to the COM-B) and context-specific theories within the studies will be presented in narrative and table form. An example table which will be used to extract the BCTs is included in online supplemental appendix 4 and a table which will be used to map the behavioural targets to the COM-B components is included in online supplemental appendix 5.

## ETHICS AND DISSEMINATION

This review will be the first step to formally identify which BCTs have been employed, what the behavioural targets were and which behavioural theories and theoretical models underpinned the interventions designed to affect MSK image reporting. This information will be used to inform an intervention design process seeking to improve the communication of imaging results. The results will be disseminated through a peer-reviewed publication, conference presentations and stakeholder events.

## Patient and public involvement

Patients with experience of receiving MSK imaging reports were invited to participate in individual sessions initially by advertising in person and using study flyers within radiology departments and general practitioner surgeries. These initial sessions sought to explore peoples' experience of imaging report communication and the ways that this could be improved, such as the setting, personnel involved and resources that would be helpful to them. These meetings highlighted clinical behaviours that were discordant with patient preference. Based on this information, it was deemed necessary to investigate the barriers to providing helpful communication of reports and the current review of whether

existing interventions had targeted these. Further patient and public involvement group sessions are planned to discuss the results of this review and to have input into the design and implementation of further work seeking to improve the communication of MSK imaging findings.

**Contributors** EK designed and wrote the protocol with planning and design input from BXWL and ABr and ABa. Studies will be screened by EK and AM. Data will be extracted by EK and AM. Data will be coded by EK and AB. AM, BXWL, ABr and Aba will review and make amendments to drafts of the manuscript and approve the final draft.

**Funding** EK is an ICA Pre-doctoral Clinical and Practitioner Academic Fellow supported by Health Education England and the National Institute for Health and Care Research. The University of Essex have provided funding for associated Article Processing Charges.

**Competing interests** None declared.

**Patient and public involvement** Patients and/or the public were involved in the design, or conduct, or reporting, or dissemination plans of this research. Refer to the Methods section for further details.

**Patient consent for publication** Not applicable.

**Provenance and peer review** Not commissioned; externally peer reviewed.

**ORCID iDs**
Edward Kirby http://orcid.org/0000-0001-8105-6505
Bernard X W Liew http://orcid.org/0000-0002-7057-7548

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
