## [Reviewer comments · BMJ Open]

ARTICLE DETAILS

TITLE (PROVISIONAL)	Characterising the interventions designed to affect the reporting of musculoskeletal imaging. A scoping review protocol using the Behaviour Change Wheel.
AUTHORS	Kirby, Edward; MacMillan, Andrew; X.W. Liew, Bernard; Brinkley, Andrew; Bateman, Andrew

VERSION 1 – REVIEW

REVIEWER	Kjelle, Elin Norwegian University of Science and Technology, Department of Health Sciences in Gjøvik
REVIEW RETURNED	06-Mar-2023

GENERAL COMMENTS	Thank you for the opportunity to review this protocol. It is a good protocol. However, I have a few questions about additional information I would like to see included: - Will the papers to be included be quality appraised for bias in any way? This issue is addressed in the PRISMA statement item 11: "Specify the methods used to assess risk of bias in the included studies, including details of the tool(s) used, how many reviewers assessed each study and whether they worked independently, and if applicable, details of automation tools used in the process." this would be important for the quality of the review and should be described in this protocol. If bias is not to be assessed, then this should be discussed in the protocol.- When will the searches be conducted?- Are there any limits on publication year for included studies? In addition, an open question: Have you considered including nuclear medicine imaging for MSK pain? Especially for oncology, this might be relevant.
---

REVIEWER	Costa, Nathalia The University of Queensland
REVIEW RETURNED	15-Mar-2023

GENERAL COMMENTS	Thank you for the opportunity to review this protocol. The authors have designed a sound protocol for scoping review that seeks to understand the scope of behavioural targets, intended mechanisms of action and theoretical underpinnings of interventions seeking to reduce the negative effects of musculoskeletal imaging reporting. The scoping review has the potential to make a meaningful contribution to the field of musculoskeletal pain, particularly because interventions targeting behaviour change in this field often lack a theoretical foundation. I commend the authors on their initiative – this work is not only likely to inform future interventions seeking to reduce the negative
---

	effects of musculoskeletal imaging reporting, but also highlight the importance of fostering theory-driven research approaches in musculoskeletal pain research more broadly (which is way overdue!). I only have a few minor comments to offer and I hope the authors find them useful:  1. The authors cite Witherow's review and mention that such a review identified little evidence of effectiveness for any of the interventions. In the review that the authors are proposing here, it seems that the outcomes of the included studies will not be extracted and/or described. I understand that this is a scoping review and that reporting the effectiveness of the interventions is not the aim here, but I think readers would appreciate a summary of the results reported in the included studies. What if by the time the search strategy is run a study that reported some degree of effectiveness gets published? It would be good to enable readers to match the different theories and/or mechanisms of action to the outcomes of the included studies. The authors could even interpret the data and write a summary reporting the relationships between these aspects. 2. It seems that the authors are extracting data about the mechanisms of action and the categories of theories but not the theories? What if some of the studies specifically mention a theory or a theoretical framework that informed the intervention? I think it would be important to extract this data too. 3. It might be worth considering reporting barriers and enables if the included studies report those. While I do not think binaries are always helpful, they do have the potential to provide important insights into the context in which these interventions are implemented and/or tested and how different contextual factors may influence the outcomes identified. 4. Page 6 mentions question 2 on the template but question 1 is not mentioned prior to that? 5. I encourage the authors to create a template of a table that gives the readers some insight into what the reporting of the data may look like in a table form and add such a template as an appendix.
--	---

VERSION 1 – AUTHOR RESPONSE

Reviewer 1 Comments:

An Assessment of Methodological Quality section has been added, explaining why this was not performed.

Search dates and inclusion criteria based on publication year has been added to the search strategy section.

Open question - consideration was given to the nuclear medicine imaging within the search terms, however as this study wishes to exclude serious causes of MSK pain, such as cancers, for which nuclear medicine imaging is typically used and because 'CT' is already included in the search terms, which we feel would capture the utilisation of nuclear medicine for non-serious conditions, it was decided on balance not use this term.

Reviewer 2 questions:

1. a summary of outcomes section has been added to the study characteristics data extraction template. The extracted outcome data will be referred to when discussing intervention effectiveness in the discussion.
2. A clarifying sentence has been added to the start of the last paragraph in the Data Extraction section explaining that the explicit theories will indeed be extracted.
3. Reporting the barriers and enablers would be highly beneficial. I therefore intend to perform a separate review to identify these barriers, enablers and behavioural theories identified in the qualitative literature. In addition, I have added an 'influencing factors' section to the 'study characteristics' extraction to capture any barriers, enablers, results of sub-group analysis, moderation effects and confounders found in Quantitative studies.
4. Question 2 on page 6 has been amended to say 'Item 2 on the Tidier template'.
5. two templates of the tables which will be used to display the results have been added to the appendix

VERSION 2 – REVIEW

REVIEWER	Kjelle, Elin Norwegian University of Science and Technology, Department of Health Sciences in Gjøvik
REVIEW RETURNED	03-Jul-2023
GENERAL COMMENTS	-
REVIEWER	Costa, Nathalia The University of Queensland
REVIEW RETURNED	07-Jul-2023
GENERAL COMMENTS	Thank you for addressing all the points raised previously. I hope the review process goes well, and I look forward to reading the results of this work. All the best!

VERSION 2 – AUTHOR RESPONSE